# Mechanism of Marinobufagenin-Induced Hyperpermeability of Human Brain Microvascular Endothelial Cell Monolayer: A Potential Pathogenesis of Seizure in Preeclampsia

**DOI:** 10.3390/cells13211800

**Published:** 2024-10-30

**Authors:** Ahmed F. Pantho, Manisha Singh, Syeda H. Afroze, Kelsey R. Kelso, Jessica C. Ehrig, Niraj Vora, Thomas J. Kuehl, Steven R. Lindheim, Mohammad N. Uddin

**Affiliations:** 1Artemis Biotechnologies LLC, Temple, TX 76502, USA; ahmed.pantho@artemisbiotech.org (A.F.P.); afrozesyeda751@gmail.com (S.H.A.); tjkuehl@peoplepc.edu (T.J.K.); 2Neonatal and Perinatal Medicine, Baylor Scott & White Hospital, Temple, TX 75182, USA; manisha.singh@bswhealth.org (M.S.); niraj.vora@bswhealth.org (N.V.); steven.lindheim@bswhealth.org (S.R.L.); 3Obstetrics & Gynecology, Baylor Scott & White Hospital, Temple, TX 75182, USA; kelsey.kelso@bswhealth.org (K.R.K.); jessica.ehrig@bswhealth.org (J.C.E.); 4Texas A&M University College of Medicine, College Station, TX 77807, USA

**Keywords:** marinobufagenin, HBMEC, MAPK, apoptosis, tight junction proteins

## Abstract

Preeclampsia (preE) is a hypertensive disorder in pregnancies. It is the third leading cause of mortality among pregnant women and fetuses worldwide, and there is much we have yet to learn about its pathophysiology. One complication includes cerebral edema, which causes a breach of the blood–brain barrier (BBB). Urinary marinobufagenin (MBG) is elevated in a preE rat model prior to developing hypertension and proteinuria. We investigated what effect MBG has on the endothelial cell permeability of the BBB. Human brain microvascular endothelial cells (HBMECs) were utilized to examine the permeability caused by MBG. The phosphorylation of ERK1/2, Jnk, p38, and Src was evaluated after the treatment with MBG. Apoptosis was evaluated by examining caspase 3/7. MBG ≥ 1 nM inhibited the proliferation of HBMECs by 46–50%. MBG induced monolayer permeability, causing a decrease in the phosphorylation of ERK1/2 and the activated phosphorylation of Jnk, p38, and Src. MBG increased the caspase 3/7 expression, indicating the activation of apoptosis. Apoptotic signaling or the disruption of endothelia tight junction proteins was not observed when using the p38 inhibitor as a pretreatment in MBG-treated cells. The MBG-induced enhancement of the HBMEC monolayer permeability occurs by the downregulation of ERK1/2, the activation of Jnk, p38, Src, and apoptosis, resulting in the cleavage of tight junction proteins, and are attenuated by p38 inhibition.

## 1. Introduction

Preeclampsia (preE) is defined by the emergence of hypertension and proteinuria after 20 weeks of pregnancy, frequently accompanied by notable edema and intrauterine growth restriction. It ranks as the third most common cause of morbidity and mortality among mothers and fetuses, contributing to an estimated 60,000 maternal deaths globally each year. Neurological issues associated with preE can include transient deficits such as cortical blindness, aphasia, weakness, paralysis, and cerebrovascular accidents [1]. It is estimated that preE is responsible for nearly half of all reversible strokes related to pregnancy [2,3]. The neurological effects of preE can be explained by two primary hypotheses: (1) vasospasm resulting from severe hypertension, which can lead to cytotoxic edema, ischemia, and infarction [4]; and (2) hypertensive events that surpass cerebrovascular autoregulation, causing a cycle of vasoconstriction and forced vasodilation. There is substantial evidence supporting the notion that preE is linked to endothelial dysfunction and oxidative stress [2]. Consequently, the integrity of endothelial cell tight junctions is compromised, resulting in plasma leakage and the formation of cerebral edema [5,6].

Obstetricians have long suspected the presence of a “vascular leak” syndrome in preeclampsia (preE) [7,8,9]. This suspicion was supported by studies that showed a higher rate of Evan’s blue dye disappearance from the circulation in preE patients compared to normal pregnancies, indicating increased vascular permeability [10,11]. In experimental models, a condition resembling human preE can be induced in pregnant rats by administering weekly desoxycorticosterone acetate injections and replacing their drinking water with saline [4]. A similar syndrome can also be triggered by daily injections of marinobufagenin (MBG) in early pregnancy [4]. We have evaluated vascular leakage in this rodent “preE” model and observed greater leakage in preE rats compared to normal pregnant and non-pregnant female rats [12]. The vascular integrity was assessed by observing single post-capillary venules in the omentum, using fluorescein isothiocyanate-albumin (FITC albumin) injections to detect any extravasation of intravascular fluid [12]. Moreover, the intravenous injection of 200 nmol MBG into control rats caused the extravasation of labeled albumin within minutes. Subsequent studies on rat lung endothelial cell monolayers incubated with increasing doses of MBG demonstrated hyperpermeability, linked to the disruption of tight junctions between cells [13]. Vascular endothelial (VE)-cadherin, a transmembrane adhesive protein, plays a crucial role in forming endothelial adherens junctions, which are key regulators of paracellular permeability in microvascular endothelium [14,15,16,17]. Changes in VE-cadherin are linked to the increased permeability of the endothelial monolayer [18].

While increased vascular permeability is recognized as a key factor in the pathogenesis of preeclampsia (preE) [6,18], the role of circulating factors in causing the endothelial barrier dysfunction remains unclear. In this research, human brain microvascular endothelial cells (HBMECs) were used to investigate whether marinobufagenin (MBG) plays a role in the increased vascular permeability seen in preE. The study focused on examining the effects of MBG on the permeability of HBMEC monolayers, as well as its influence on the phosphorylation of ERK, Jnk, p38, and Src. Additionally, early apoptotic markers, including caspases 3/7 activity and annexin-V staining, were analyzed to assess the impact of MBG. Additionally, we looked at how MBG influences endothelial adherens junction proteins within the HBMEC monolayer. Furthermore, we evaluated whether inhibiting p38, Jnk, and Src could prevent the increase in microvascular permeability. Our hypothesis was that MBG enhances endothelial cell permeability by upregulating p38, Src, and caspase-mediated apoptosis, as well as by disrupting endothelial cell tight junctions. Moreover, we hypothesized that the p38 inhibition attenuated the MBG-induced disruption of tight junction proteins, and thus the HBMEC monolayer hyperpermeability.

## 2. Materials and Methods

Cell culture: Human brain microvascular endothelial cells (HBMECs) obtained from ScienCell Research Laboratories (Carlsbad, CA, USA) were cultured on poly-L-lysine-coated dishes using complete endothelial cell medium (ECM) supplemented with 5% fetal bovine serum, 1% endothelial cell growth supplement, and 1% penicillin/streptomycin. The cells were maintained in an incubator (Isotemp CO_2_ incubator, Fisher Scientific, Pittsburgh, PA, USA) set to 37 °C, with 5% CO_2_ and 99% humidity.

Cell proliferation: Cell proliferation following MBG treatment was evaluated using a method previously described in earlier studies [5,19,20,21]. Human brain microvascular endothelial cells (HBMECs) were plated at a density of 7000 cells per well in 96-well plates with complete medium and allowed to adhere overnight at 37 °C. The following day, cells were serum-starved in medium with 0.5% fetal bovine serum (FBS) for 24 h, then washed twice with 1× phosphate-buffered saline (PBS). Eight replicates were treated with 10% endothelial cell medium (ECM) to stimulate proliferation, containing either DMSO (as a basal control) or varying concentrations of MBG at 0.1, 1, 10, or 100 nM. After a 48 h incubation period with the treatments, 10 µL of CellTiter 96 was then added to each of the wells, and absorbance at 490 nm was recorded using a microplate reader (Microplate Spectrophotometer, SpectraMax 3400 PC, Molecular Devices, Sunnyvale, CA, USA). The measured absorbance at this wavelength correlates directly with the number of viable cells present [5].

Cell viability: Cell viability was assessed using a method previously described in the literature [21]. This assay is based on the ability of live cells to convert the redox dye resazurin into the fluorescent compound resorufin. Human brain microvascular endothelial cells (HBMECs) were plated in 96-well plates with complete medium and allowed to adhere overnight at 37 °C. The cells were then exposed to either DMSO (serving as the basal control) or varying concentrations of marinobufagenin (MBG) at 0.1, 1, 10, or 100 nM. Following a 48 h incubation period, 20 µL of CellTiter-Blue reagent was added to each well. Absorbance readings at 520 nm were obtained using a microplate reader, with the signal intensity directly reflecting the cell viability based on the conversion of resazurin to resorufin.

Permeability: The monolayer permeability assay was performed following a method previously reported in the literature [15]. Human brain microvascular endothelial cells (HBMECs) were grown as monolayers on Costar Transwell membranes (Corning Inc., Corning, NY, USA) for 48 h. Prior to the experiment, the cells were exposed to fresh, phenol-red-free media for one hour. Subsequently, they were treated with either DMSO (as a basal control) or various concentrations of marinobufagenin (MBG) at 0.1, 1, 10, or 100 nM for a period of 6 h. In some experiments, the cells were pre-incubated for 2 h with inhibitors targeting p38 (10 µM SB 202190, Sigma, St. Louis, MO, USA), Jnk (20 µM SP 600125), or Src (20 µM, Invitrogen, Carlsbad, CA, USA) prior to treatment with either DMSO or MBG. To assess permeability, FITC-albumin was added to the luminal chamber at a concentration of 5 mg/mL for 60 min, after which samples were collected from both luminal and abluminal chambers. Samples (100 µL) from the abluminal side were analyzed using a fluorometric plate reader (excitation 400 nm/emission 505 nm). Fluorescence values were converted into albumin concentrations using a standard calibration curve. These concentrations were then used to determine the albumin permeability coefficient (Pa) according to the following formula: P_a_ = ([A]/t) × (1/A) × (V/[L])—where [A] is the concentration in the abluminal chamber (mg/mL), t is the duration in seconds, A represents the membrane area in cm^2^, V is the volume of the abluminal chamber (ml), and [L] is the concentration in the luminal chamber (mg/mL).

Effect of MBG on ERK 1/2, Jnk, p38, and Src phosphorylation: The effect of marinobufagenin (MBG) on the phosphorylation of ERK 1/2, Jnk, p38, and Src was evaluated using the Cellular Activation of Signaling (CASE) ELISA kit (SuperArray, Frederick, MD, USA) [22]. This kit employs an antibody-based detection system that enables the colorimetric quantification of both phosphorylated and total protein levels. For the assay, human brain microvascular endothelial cells (HBMECs) were cultured in 96-well plates and treated with either DMSO (as a control) or varying concentrations of MBG (0.1, 1, 10, or 100 nM) for different time points (10, 30, 60, 120, and 240 min). The measurements of the total and phosphorylated protein expression were performed following the manufacturer’s protocol.

Effect of MBG on caspase 3/7 activity: The Caspase-Glo 3/7 assay, a luminescent method designed to detect caspase 3/7 activity, is well suited for the automated high-throughput screening of apoptosis in multi-well-plate setups. In this study, the assay was conducted in a 96-well format, where cells were exposed to DMSO (as a control) or marinobufagenin (MBG) at concentrations of 0.1, 1, 10, or 100 nM for a period of 4 h. In certain experiments, cells were pre-incubated with either a p38 or Jnk inhibitor for 2 h before MBG treatment. After treatment, 100 µL of Apo-ONE Caspase 3/7 reagent (Promega, Madison, WI, USA) was added to each well, followed by a 1 h incubation at 37 °C. Luminescence readings were then obtained using a luminometer (Fluoroskan Ascent FL, Thermo Labsystems, Franklin, MA, USA), following the method described in previous studies [23].

Annexin-V staining: HBMECs were treated with DMSO (as a control) or different concentrations of marinobufagenin (MBG) at 0.1, 1, 10, or 100 nM for a 24 h period. In certain experiments, cells were pre-incubated for 2 h with a p38 inhibitor (10 µM SB 202190) or a Jnk inhibitor (20 µM SP 600125) before being exposed to 10 nM MBG for an additional 24 h. Some groups were treated solely with the p38 or Jnk inhibitor. After these treatments, cells were incubated for 30 min with a 1:50 dilution of biotinylated annexin-V (Roche, Indianapolis, IN, USA) in an incubation buffer. Following incubation, cells were fixed with 4% paraformaldehyde (PFA) for 10 min at room temperature. They were then treated with a 1:200 dilution of Cy3-labeled streptavidin (GE Healthcare UK Ltd., Buckinghamshire, England) for 30–60 min at room temperature. After staining, the cells were mounted on microscope slides with coverslips, along with the nuclear marker DAPI (4′,6′-diamidino-2-phenylindole). Visualization was performed using an Olympus FluoView FV 300 confocal laser-scanning microscope.

Effect of MBG on endothelial tight junction proteins: HBMECs were grown on poly-L-lysine-coated glass chamber slides and treated with MBG at doses of 0.1, 1, 10, or 100 nM for a 24 h period. A separate control group received DMSO (basal control). Additionally, certain cells were pre-incubated for 2 h with a p38 inhibitor (10 µM SB 202190) prior to exposure to 10 nM MBG for 24 h. After the MBG treatments, the cells were rinsed with PBS and fixed in 4% paraformaldehyde. Following multiple wash steps, the cells were permeabilized with Triton X-100 and blocked to minimize nonspecific binding. The cells were then incubated overnight at 4 °C with primary antibodies targeting ZO-1, Occludin, Claudin-1, or E-Cadherin. After rinsing with PBS, the cells were treated with an FITC-labeled secondary antibody for 1 h. Further washing was performed, and the cells were mounted in an antifade medium containing the nuclear dye DAPI. Imaging was conducted using an Olympus FluoView FV 300 confocal laser-scanning microscope, with appropriate filters set for FITC and DAPI detection.

Statistical analysis: Data are presented as the mean ± standard error of the mean (S.E.M.). To compare MBG-treated groups with the control group treated with DMSO, one-way ANOVA was employed, followed by Tukey’s post hoc test for multiple comparisons. A *p*-value of ≤0.05 was considered statistically significant.

## 3. Results

### 3.1. MBG Inhibited HBMEC Proliferation

As shown in Figure 1, treatment with MBG significantly decreased HBMEC proliferation compared to the DMSO (control) group, with 1, 10, and 100 nM concentrations causing 66%, 70%, and 70% inhibition, respectively (*p* < 0.05 for all). In contrast, the 0.1 nM MBG treatment did not affect HBMEC proliferation. The anti-proliferative effect of MBG was not attributed to cytotoxicity, as demonstrated by cell viability assays [13]. 

### 3.2. MBG Increased Monolayer Permeability

Treatment with MBG at concentrations of 1, 10, and 100 nM led to a significant increase in the HBMEC monolayer permeability, of approximately 1.5-fold (*p* < 0.05 for all), as illustrated in Figure 2. The 0.1 nM MBG treatment did not influence permeability. To determine whether the observed hyperpermeability was mediated through the p38, Jnk, or Src pathways, HBMECs were pretreated with specific inhibitors for these kinases prior to exposure to DMSO (control) or MBG. The p38 and Src inhibitors effectively reduced the hyperpermeability caused by 1, 10, or 100 nM MBG (*p* < 0.05 for each), while the Jnk inhibitor did not demonstrate a similar effect.

### 3.3. MBG Downregulated ERK 1/2 Phosphorylation

To determine whether the observed decrease in proliferation and increased monolayer permeability were linked to reduced ERK1/2 activity, the phosphorylation levels of ERK1/2 were measured in HBMECs. As shown in Figure 3, treatment with 1, 10, and 100 nM MBG led to a significant decline in the ratio of phosphorylated ERK1/2 to total ERK1/2 compared to the control cells, with decreases of 80%, 80%, and 72% at 10, 30, and 60 min, respectively (*p* < 0.05 for comparisons between the DMSO control and the MBG treatments). The ratios of phosphorylated to total ERK1/2 at 2 and 4 h did not show significant differences from the 1 h time point [13]. Furthermore, MBG did not affect the total ERK protein levels, suggesting that the observed changes were specifically related to decreased phosphorylation.

### 3.4. MBG Increased Phosphorylation of Jnk, p38, and Src

In HBMECs treated with 1, 10, and 100 nM MBG, there was a significant rise in the ratio of phosphorylated Jnk to total Jnk compared to the control, with increases of 50%, 56%, and 56% observed at 10, 30, and 60 min, respectively (*p* < 0.05 for comparisons between the DMSO control and MBG treatments) (Figure 4A). Similarly, MBG at these concentrations significantly enhanced the ratio of phosphorylated p38 to total p38, demonstrating increases of 70%, 71%, and 71% at 10, 30, and 60 min, respectively (*p* < 0.05 for comparisons between DMSO control and MBG treatment) (Figure 4B). Furthermore, the MBG treatment resulted in a substantial increase in the ratio of phosphorylated Src to total Src, with rises of 70%, 71%, and 71% at 10, 30, and 60 min, respectively (*p* < 0.05 for comparisons between DMSO control and MBG treatments) (Figure 4C). The ratios of phosphorylated to total proteins at 2 and 4 h did not significantly differ from those measured at 1 h for all conditions [13]. Importantly, MBG did not influence the total levels of Jnk, p38, or Src proteins, indicating that the observed increases in ratios were specifically due to enhanced phosphorylation.

### 3.5. Activation of Apoptosis by MBG in HBMECs Was Attenuated by p38 Inhibition

Apoptosis was assessed using caspase 3/7 activity and Annexin-V staining. As depicted in Figure 5A, treatment with 1, 10, and 100 nM MBG led to a significant (~2.0-fold) increase in caspase 3/7 activity in HBMECs after 3 h, compared to baseline levels (*p* < 0.05 for comparisons between DMSO (basal) and 1, 10, or 100 nM MBG). Treatment with 0.1 nM MBG did not affect caspase 3/7 activity (Figure 5A). Pretreatment with a p38 inhibitor effectively blocked the MBG-induced caspase 3/7 activation (*p* < 0.05), whereas the Jnk inhibitor did not impact this effect. Annexin-V staining also showed increased apoptotic cells in HBMECs treated with 1, 10, and 100 nM MBG, but not with 0.1 nM MBG. Notably, apoptosis was not observed in cells treated with the p38 or Jnk inhibitors alone. These findings suggest that the apoptosis observed was mediated through the p38 signaling pathway (Figure 5B).

### 3.6. MBG Caused the Disruption of Endothelial Adherens Tight Junctions That Was Attenuated by p38 Inhibition

In HBMECs treated with DMSO, the tight junctions were well preserved, exhibiting clear and consistent staining for ZO-1, Occludin, Claudin-1, and E-Cadherin at the cell–cell interfaces (Figure 6), suggesting a robust cell barrier. In contrast, exposure to 1, 10, and 100 nM MBG resulted in the significant disruption of these tight junctions, characterized by irregular and dispersed staining patterns for ZO-1, Occludin, Claudin-1, and E-Cadherin. Notably, treatment with 0.1 nM MBG did not impact the integrity of the tight junctions. Additionally, the application of a p38 inhibitor was effective in reducing the disruption of these junction proteins induced by MBG (Figure 6).

## 4. Discussion

Marinobufagenin (MBG) is a circulating steroid [24] that is excreted unchanged in the urine [25]. It belongs to a class of compounds known as bufodienolides, named after their discovery in the skin and venom of the toad species *Bufo marinus* [24]. Bufodienolides, along with their closely related counterparts, the cardenolides, form two categories of cardioactive steroids collectively referred to as “cardiac glycosides” [24]. These compounds are characterized by their ability to inhibit the enzyme Na^+^/K^+^ ATPase, which underlies their physiological and pathological effects. Notably, they function as cardiac inotropes, promote vasoconstriction that can contribute to hypertension, and induce natriuresis [26]. The main structural distinction between these two groups is that cardenolides feature a five-membered lactone ring, whereas bufodienolides possess a doubly unsaturated six-membered lactone ring. Mechanistically, bufodienolides preferentially target the α-1 isoform of Na^+^/K^+^ ATPase, while cardenolides primarily interact with the α-2 and α-3 isoforms.

Marinobufagenin (MBG) has been shown to affect vascular permeability in post-capillary venules within the splanchnic circulation of rats with preeclampsia [12]. In these studies, the administration of MBG resulted in a rapid escape of labeled albumin into the extravascular space [12]. In vitro investigations involving endothelial cell monolayers revealed comparable outcomes, as MBG increased the permeability of the endothelial cell layer by inhibiting cell proliferation in rat lung microvascular endothelial cells. This bufodienolide was found to decrease ERK 1/2 phosphorylation while enhancing the phosphorylation of Jnk and p38. Moreover, MBG elevated the levels of caspases 3, 7, 8, and 9, suggesting that it promotes apoptotic activity. Additionally, MBG was associated with the disruption of endothelial cell junctions [27].

Clinical studies comparing the cerebrovascular responses in preE pregnant women indicate increased cerebral perfusion pressures and decreased vasodilation compared to normotensive pregnant women [28,29]. Such changes in vascular pressure may dysregulate endothelial cell tight junctions and adversely impact the blood–brain barrier (BBB) integrity. Tokuda et al., employing an in vitro model utilizing renal epithelial cells, reported that hydrostatic pressure increases transepithelial resistance in endothelial cells, rapidly changes the localization of the tight junction protein claudin-1 and increases cellular ion transport [30]. Sustained pressure on the endothelium, as in preE, may facilitate permeability in numerous microvascular beds. In the brain, this may result in neuronal exposure to inflammatory and potentially cytotoxic compounds. Thus, a potential explanation for the reversible cortical blindness, which occasionally occurs in preeclamptic women associated with gray-white matter edema, may be a specific manifestation of hypertension-induced barrier permeability [31]. PreE-induced BBB changes are strongly supported by our data on the effects of MBG incubation on monolayers of HBMECs. We found that MBG raised the monolayer permeability of HBMECs. To explore the mechanisms behind altered BBB permeability, we investigated the impact of MBG on HBMECs. Our findings revealed that MBG treatment reduced HBMEC proliferation, paralleling its effects observed in cytotrophoblast (CTB) cell proliferation [21,27]. Additionally, MBG increased the permeability of the HBMEC monolayer. We specifically examined the roles of the p38 and Src signaling pathways in contributing to MBG-induced BBB hyperpermeability. By inhibiting p38 and Src activation following MBG exposure, we observed a significant reduction in microvascular hyperpermeability (Figure 3). Src-mediated phosphorylation is critical for BBB disruption, as highlighted by Farkas et al. [32], and Takenaga et al. demonstrated that Src inhibition can mitigate BBB permeability increases following transient focal cerebral ischemia [33]. Our study also found that MBG triggered apoptosis in HBMECs, as evidenced by increased caspase 3/7 activity (Figure 5A) and positive annexin-V staining (Figure 5B), effects consistent with previous reports in CTB cells [27].

Our results demonstrate that MBG treatment disrupts tight junctions in endothelial cells, an effect that can be alleviated by using a p38 inhibitor (Figure 6). This suggests that the hyperpermeability observed in endothelial cells is mediated by the activation of caspase-3/7, which cleaves tight junction proteins and leads to cell detachment. These findings are consistent with previous studies conducted in similar experimental systems [34,35,36]. It is well established that p38 MAPK plays a critical role in regulating the actin cytoskeleton of vascular endothelial cells, as well as their tight junctions and permeability [10,11,37,38,39,40]. Recent research indicates that caspase-3, a key enzyme in the apoptotic signaling pathway, is responsible for cleaving both ZO-1 and occludin in epithelial cells [41]. In these cells, caspase-3 cleaves the C-terminal cytoplasmic domain of occludin, resulting in a modified form that can no longer bind to the cytoplasmic adapter protein ZO-1, thus disrupting its association with the actin cytoskeleton. Claudins, which are also the vital components of tight junctions, interact with ZO-1. Consequently, the structural integrity of the tight junctions is compromised, breaking their linkage to the actin cytoskeleton and other proteins associated with ZO-1, ZO-2, and ZO-3 [41]. Additionally, MBG has been shown to activate endogenous caspase-3 [13,27].

## 5. Conclusions

The inclusion of MBG in the incubation medium of HBMEC monolayers results in hyperpermeability. This was a consequence of alterations in the activities of components of the MAPK signaling complex and in the upregulation of apoptosis. We observed that MBG reduced ERK1/2 phosphorylation while increasing the phosphorylation of p38 and Src. Additionally, the hyperpermeability of HBMEC monolayers and apoptotic signaling induced by MBG were inhibited by p38 inhibition, but not by Jnk inhibition. These data support the idea that MBG significantly contributes to the neurological complications commonly observed in human preeclampsia. As shown in Working Model in Figure 7, the volume expansion during pregnancy stimulates the secretion and elaboration of MBG in the circulation. The elevated MBG either by interfering with Na^+^-K^+^-ATPase or interfering with other cell surface receptor causes the activation of p38 MAPK. The activated p38 MAPK directly acts on the actin cytoskeleton and thus causes the disruption of brain tight junction proteins [38,39,40]. Alternatively, the activated p38 triggers the apoptotic signaling which in turns causes the disruption of brain tight junction proteins [41]. The disruption of tight junction proteins leads to a vascular leak syndrome, contributing to increased BBB hyperpermeability, which can result in cerebral edema, a characteristic feature of preE. In both in vitro systems and in vivo models, it has been shown that a p38 inhibitor, which targets a downstream element of this signaling pathway, can reduce MBG-induced apoptotic signaling [8,12,13,21,27]. Therapeutically targeting MBG signaling, particularly the p38 MAPK pathway, could offer potential treatment strategies for preE [12,13,21,27].

Our research provides insights into the pathophysiology of endothelial barrier function in pre-eclampsia in relation to cerebral vascularity [42]. This is important in understanding the ways in which pre-eclampsia causes cerebral insults. Cerebral problems, like cerebral edema, raised the intracranial pressure, and stroke constitute around 40 percent of deaths from pre-eclampsia [42]. Plasma volume during pregnancy increases significantly and maintaining endothelial integrity is essential to avoiding leakage into the extravascular compartment [43]. Consequently, the vascular endothelium is essential to helping maintain vascular permeability and research on endothelial barriers and junctions can provide valuable information to further managing pre-eclampsia and helping treat it [43]. In a previous study, Ing et al. also showed that marinobufagenin regulates the permeability and gene expression of brain endothelial cells [44].

## 6. Clinical Significance

Recent evidence strongly indicates that MBG plays a significant role in preeclampsia (preE) [45,46]. Since the 1980s, cardiotonic steroids (CTSs), also known as “endogenous digoxin-like factors” (EDLFs), have been recognized for their substantial increase during pregnancy-related hypertension and preE [47,48,49]. Recently, MBG has been identified as a key digoxin-like factor, with its levels markedly elevated in preE compared to endogenous ouabain (EO) [50]. We developed a highly specific ELISA for MBG [51], which showed a 5-fold increase in serum MBG and a 4-fold increase in urinary MBG levels in patients with preE compared to normotensive pregnant women [45]. Notably, plasma MBG levels rise in preE patients, while EO levels remain unchanged [46]. This research is particularly innovative as it provides in vitro data on how MBG affects blood–brain barrier (BBB) permeability in the context of preE. The pathological effects induced by MBG may be targeted to develop treatments for preE during early human pregnancy.

## Figures and Tables

**Figure 1 cells-13-01800-f001:**
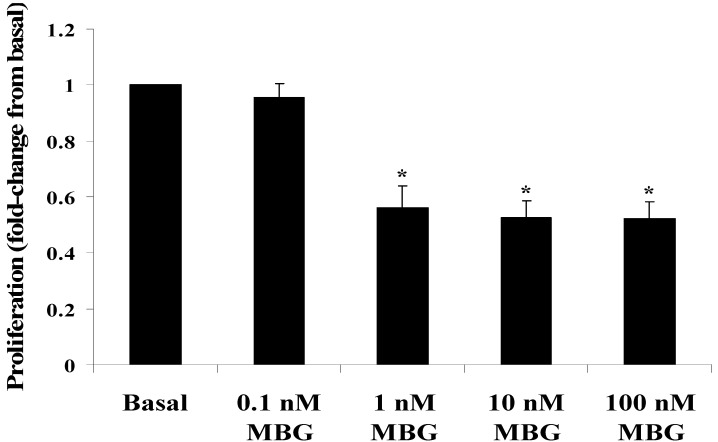
Evaluation of marinobufagenin (MBG) effect on cell proliferation. Cell proliferation (y axis) is presented as the fold-change relative to baseline values. The significant inhibition of proliferation was observed in cells treated with 1, 10, and 100 nM MBG compared to those treated with DMSO (basal; * *p* < 0.05 for all MBG concentrations). The 0.1 nM MBG treatment has no effect on cell proliferation. Data are shown as mean ± SEM (n = 5, with 8 replicates per group).

**Figure 2 cells-13-01800-f002:**
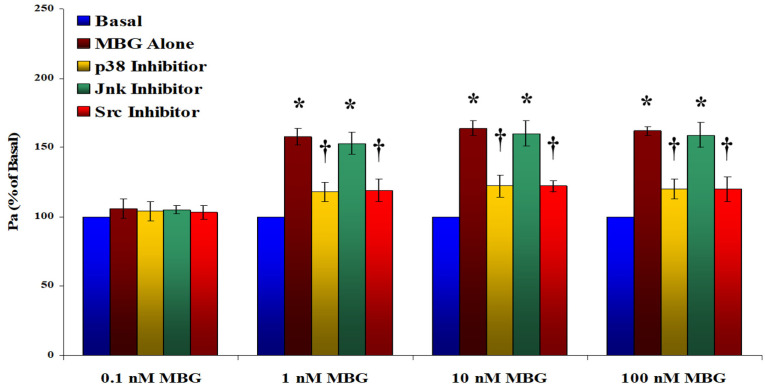
The impact of marinobufagenin (MBG) on the permeability of the human brain microvascular endothelial cell (HBMEC) monolayers. The data indicate changes in the permeability coefficient of albumin (Pa) as a percentage of baseline values on the y axis. The 0.1, 10, and 100 nM MBG treatment significantly increased HBMEC monolayer permeability within 6 h (* *p* <0.05 for each treatment). The p38 or Src inhibitors’ pretreatment significantly attenuated the MBG-induced increase in permeability († *p* <0.05 for each inhibitor). Data are shown as the mean ± SEM (n = 5, with 8 replicates for each group).

**Figure 3 cells-13-01800-f003:**
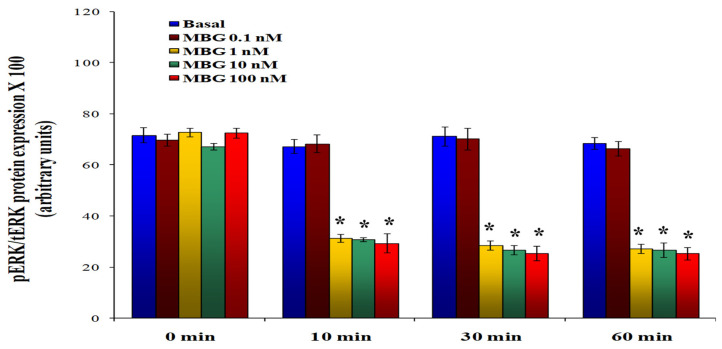
Assessment of marinobufagenin (MBG) treatment on ERK phosphorylation in human brain microvascular endothelial cells (HBMECs). The data are interpreted with the ratio of phosphorylated ERK1/2 (pERK) to total ERK1/2 (tERK). The ERK1/2 phosphorylation was statistically significantly reduced in cells treated with MBG compared to the basal control at all time points (* *p* < 0.05 basal vs. 1, 10, and 100 nM MBG treatments). The 0.1 nM MBG treatment was ineffective on ERK phosphorylation. The data are illustrated as mean ± SEM (n = 6, with 4 replicates per condition).

**Figure 4 cells-13-01800-f004:**
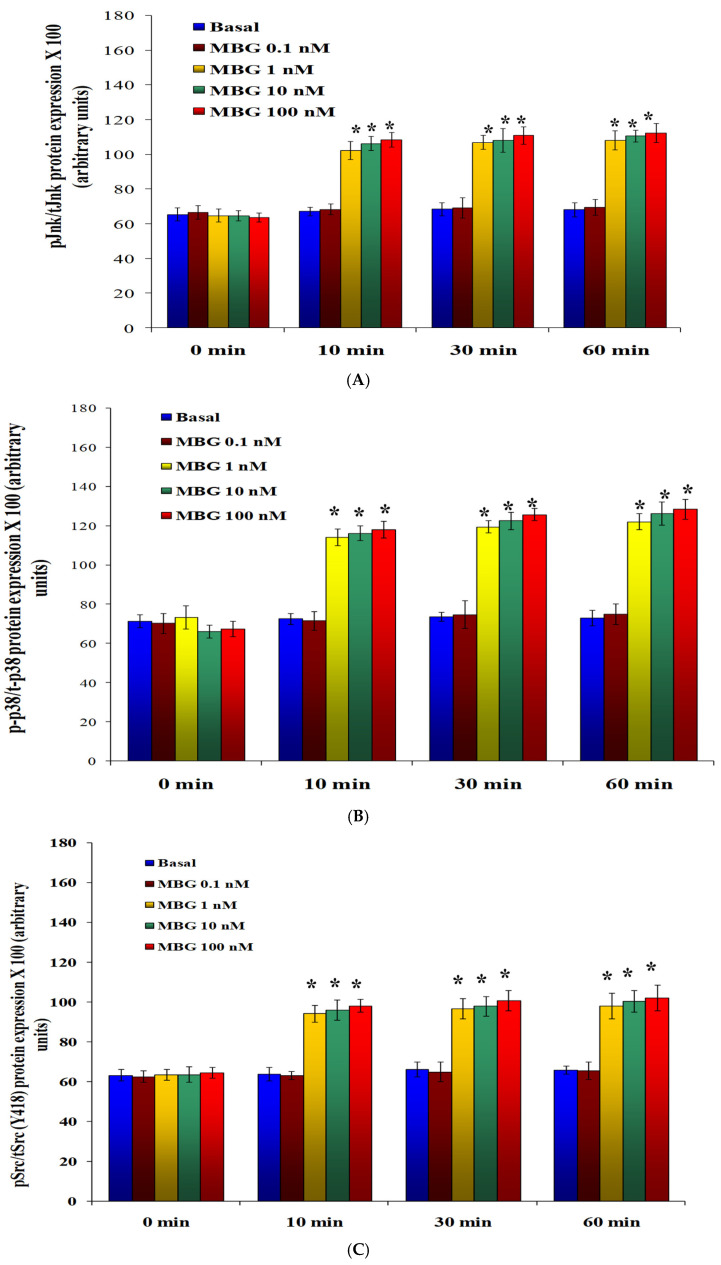
(**A**) The impact of marinobufagenin (MBG) on Jnk phosphorylation in human brain microvascular endothelial cells (HBMECs). The data were assessed by the ratio of phosphorylated Jnk (pJnk) to total Jnk (tJnk). This ratio was significantly higher in cells treated with 1, 10, and 100 nM MBG compared to the basal control at all time points measured (10, 30, and 60 min) (* *p* < 0.05 basal vs. 1, 10, and 100 nM MBG). The 0.1 nM MBG did not have any effect on Jnk phosphorylation. Results are expressed as mean ± SEM (n = 6, with 4 replicates in each time point). (**B**) The assessment of marinobufagenin (MBG) on p38 phosphorylation in human brain microvascular endothelial cells (HBMECs). The phosphorylation of p38 (p-p38) was notably elevated in HBMECs treated with 1, 10, and 100 nM MBG compared to the basal control at all time points measured (10, 30, and 60 min) (* *p* < 0.05 basal vs. 1, 10, and 100 nM MBG). The data are presented as mean ± SEM (n = 6, with 4 replicates in each time point. The MBG in a concentration of 0.1 nM MBG had no effect on p38 phosphorylation. (**C**) The effect of marinobufagenin (MBG) on Src phosphorylation in human brain microvascular endothelial cell (HBMECs). The ratio of phosphorylated Src (pSrc) to total Src (tSrc) was significantly higher in HBMECs treated with 1, 10, and 100 nM MBG compared to basal at all assessed points (10, 30, and 60 min) (* *p* < 0.05 basal vs. MBG-treated cells). Results are expressed as mean ± SEM (n = 6, with 4 replicates per group).

**Figure 5 cells-13-01800-f005:**
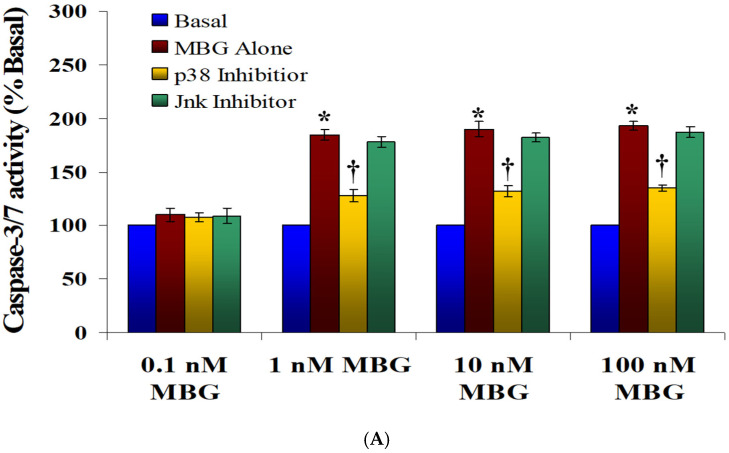
(**A**) The evaluation of marinobufagenin (MBG) on caspase 3/7 activation in human brain microvascular endothelial cells (HBMECs). The caspase 3/7 activation is shown by the percentage increase in caspase activity compared to the basal control. The MBG treatment at concentrations of 1, 10, and 100 nM statistically significantly elevated caspase 3/7 activity compared to cells treated with DMSO (* *p* < 0.05; basal and MBG-treated cells). The results are expressed as mean ± SEM (n = 5, with 8 replicates per group). Treatment with 0.1 nM MBG did not affect caspase 3/7 activity. Moreover, pretreatment with a p38 inhibitor reduced caspase 3/7 activation († *p* < 0.05), while a Jnk inhibitor did not show a significant change. (**B**) The evaluation of marinobufagenin (MBG) on apoptosis in human brain microvascular endothelial cells (HBMECs). Apoptosis was assessed using Annexin-V staining, with green indicating Annexin-V positive (apoptotic) cells and blue representing DAPI-stained nuclei. These images, captured at 60× magnification, revealed apoptosis in cells treated with 1, 10, and 100 nM MBG, whereas 0.1 nM MBG did not induce apoptosis. Pretreatment with a p38 inhibitor effectively prevented apoptosis, while a Jnk inhibitor did not have a significant impact. Neither Jnk nor p38 inhibitors alone influenced the apoptotic signaling as determined by Annexin-V staining.

**Figure 6 cells-13-01800-f006:**
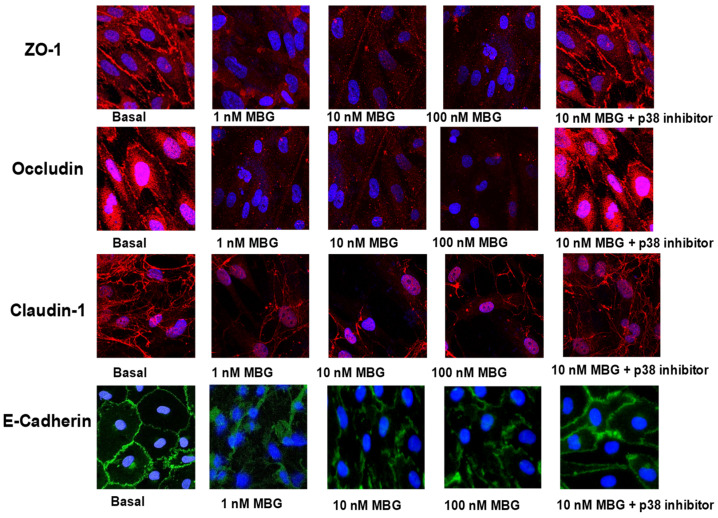
The assessment of marinobufagenin (MBG) on tight junction proteins in human brain microvascular endothelial cell (HBMECs). The immunofluorescence images of tight junction proteins in HBMECs, including ZO-1, Occludin, Claudin-1, and E-Cadherin, reveal the following: In control (basal) cells (treated with DMSO), these proteins are clearly visible with strong and continuous staining at the cell junctions, indicating intact tight junctions. However, MBG treatment disrupts these junctions. Specifically, cells treated with 1, 10, and 100 nM MBG show disrupted staining patterns with scattered ZO-1, Occludin, Claudin-1 (all red), and E-Cadherin (green). The 0.1 nM MBG treatment does not impact tight junction integrity. Furthermore, pretreatment with a p38 inhibitor effectively mitigates the MBG-induced disruption of junction proteins.

**Figure 7 cells-13-01800-f007:**
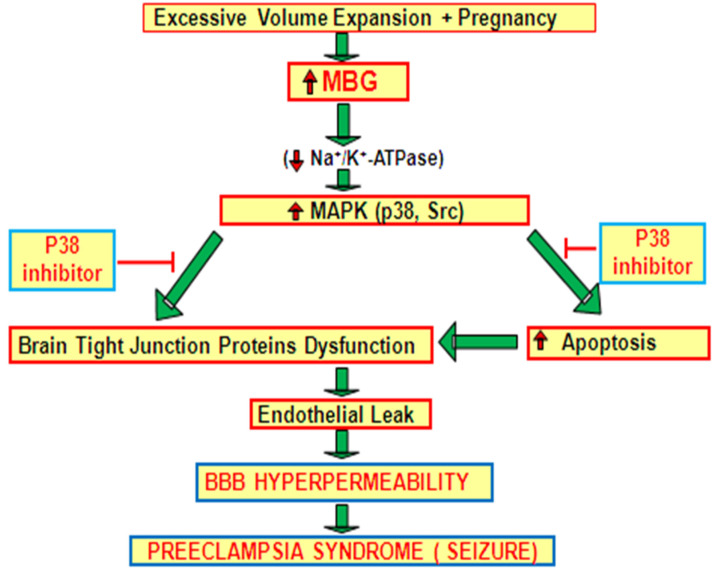
Working model.

## Data Availability

Not applicable.

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
