# Peer review of "Mechanism of Marinobufagenin-Induced Hyperpermeability of Human Brain Microvascular Endothelial Cell Monolayer: A Potential Pathogenesis of Seizure in Preeclampsia"

_cells, 2024, doi:10.3390/cells13211800_

Round 1

Reviewer 1 Report

Comments and Suggestions for Authors

The paper by dr. Pantho and colleagues reports a series of studies demonstrating that marinobufagenin may affect the permeability of human brain endothelia using a cell line of human brain microvascular endothelial cells. The Authors also provide evidence suggesting the mechanisms and the intracellular pathways of this effect potentially participating to preeclampsia pathogenesis. Although the experiments are straightforward and the data convincing, I have some problems with the quality of presentation. The manuscript looks more like a preliminary draft than a final version of a scientific manuscript.

In detail:

1. Line 13-14: I think preeclampsia might be considered the third cause of mortality not in general but among pregnant women and fetuses; and not in the U.S.A. but worldwide.

2. Line 17: marinobufagenin has been abbreviated with MBG on the line above.

3. Line 16-17: An animal model is indicated here, but it is not clear if and how it was used. All the data reported in following lines seem to refer to the HBMEC model.

4. Line 34-35: The data about morbidity and mortality seem to derive from reference [1], but in fact they are in a reference cited by [1], which reports data about maternal (and not both maternal and fetal) mortality (and not both morbidity and mortality). In addition, these data refer to the year 2000, and probably something has changed during the last 24 years.

5. There is extensive misuse of abbreviations: IUGR, DOCA, NPM, CB, NPC, PDS, and NP are never used abbreviations in the text, and they should not be introduced. Then, it is difficult (at least for me) to understand the meaning of NPM, CB, and NPC. In addition, “MBG” is used several times in the introduction, but the abbreviation is introduced only on line 71. “preE”, instead, is introduced at the beginning of introduction, but then it is repeated on lines 47 and 69. These observations are only for the introduction, I suggest revising the whole manuscript with some attention. I know that these observations may seem pedantic, but there are rules in scientific communication that should be observed and that testify the accuracy and the attention made in the preparation of a manuscript.

6. Line 39-45: Two hypotheses are introduced here: they refer to hypertension-caused vasospasm and altered cerebrovascular regulation. Then endothelial dysfunction is introduced: is this a third hypothesis? What is the link with hypotheses 1 and 2?

7. Line 62-63: The sentence beginning with “Increased …” is a repetition of lines 42-45.

8. MBG is introduced without an explanation and without an apparent reason. To also allow non-experts of preeclampsia to read and understand this research (I am not a preeclampsia expert either), I think it would be necessary to say that MBG is implicated in the pathogenesis of the disease and provide some references supporting this evidence. Overall, the introduction needs substantial rearrangement.

9. Lines 153, 154, 168: You do not need to repeat the source (Sigma or Invitrogen) every time you mention the p38 or the Jnk inhibitor.

10. Paragraph 3.1 and fig. 1: The treatment with DMSO is called “vehicle” in the text and “basal” in figure 1: please be consistent. The statistical significance is reported as p<0.001 in the text and p<0.05 in the figure legend. In the figure legend, “*p<0.05 for all MBG concentrations” is not clear (one MBG concentration does not have a significant effect): you should indicate with which value the comparison was made. In this case, it should be “*p<0.05 (or p<0.001) vs Basal (or Vehicle)”. The sentence “No significant effect …” is not necessary and should be removed. The legend of y axis says “fold-change from basal”, while in other figures it is “% of Basal”. Please be consistent.

11. Fig. 2: Similar to fig.1, the text says p<0.001, while the figure legend says p<0.05. Then, “*p<0.05 for each MBG concentrations” does not mean much. It should be replaced with “*p<0.05 (or p<0.001) vs Basal (or Vehicle)”. The sentence “The 0.1 nM MBG …” is not necessary and should be removed. The sentence of line 262-263 is not clear: what does the symbol † indicate? Is it p<0.05 vs Basal? Or vs MBG (“for each concentration” is not explicative)? Is this sentence saying that the yellow and the red histograms represent values that are significantly different from both “Basal” and the corresponding MBG concentration?

12. Fig. 3: see the comments to figures 1 and 2. In addition, those on the y axis are not “arbitrary units”, since each number represents the result of a ratio, and a ratio does not have a unit. But, why in this figure (and also figures 4A-C) data are not expressed as % of control, as in the other figures? The sentence “Data are shown as mean ± SEM”, with the indication of n and replicates, is not present in all figure legends. Please be consistent. Please also check all the figure legends for comments to figures 1 – 3.

13. Line 205, “control cells”: different words are used to identify controls all over the text: Basal, Vehicle, DMSO-treated, Control, Baseline. Please be consistent.

14. Line 296: please specify p<0.05 vs …?

15. Figures 5B and 6: Please provide a scale bar.

16. Fig. 6: an FITC-conjugated secondary antibody has been used and observations have been made with a confocal microscope equipped with FITC filters (methods). Since FITC emits a green fluorescence, where does the red color of ZO-1, occludin, and claudin-1 come from?

17. Lines 355 and 365: what are the CTB cells?

18. Acknowledgments: the Authors thank dr. Mor for the CTB cell line, but these cells do not seem to have been used in these studies.

19. Discussion: as in the introduction, some basic information is missing, at least in my opinion, about the evidence that MBG is involved in the pathogenesis of preE. This is an essential prerequisite to justify these studies and to propose mechanistic models.

Author Response

The paper by dr. Pantho and colleagues reports a series of studies demonstrating that marinobufagenin may affect the permeability of human brain endothelia using a cell line of human brain microvascular endothelial cells. The Authors also provide evidence suggesting the mechanisms and the intracellular pathways of this effect potentially participating to preeclampsia pathogenesis. Although the experiments are straightforward and the data convincing, I have some problems with the quality of presentation. The manuscript looks more like a preliminary draft than a final version of a scientific manuscript.

In detail:

  1. Line 13-14: I think preeclampsia might be considered the third cause of mortality not in general but among pregnant women and fetuses; and not in the U.S.A. but worldwide.

Response: It has been taken care of.

  1. Line 17: marinobufagenin has been abbreviated with MBG on the line above.

Response: It has been taken care of.

  1. Line 16-17: An animal model is indicated here, but it is not clear if and how it was used. All the data reported in following lines seem to refer to the HBMEC model.

Response: Urinary marinobufagenin (MBG) is elevated in a preE rat model prior to developing hypertension and proteinuria25 (reference 25).

  1. Line 34-35: The data about morbidity and mortality seem to derive from reference [1], but in fact they are in a reference cited by [1], which reports data about maternal (and not both maternal and fetal) mortality (and not both morbidity and mortality). In addition, these data refer to the year 2000, and probably something has changed during the last 24 years.

Response: A new reference has been replaced with the old reference. “Fishel Bartal M, Sibai BM. Eclampsia in the 21st century. Am J Obstet Gynecol 2022, 226(2S): S1237-S1253.”

  1. There is extensive misuse of abbreviations: IUGR, DOCA, NPM, CB, NPC, PDS, and NP are never used abbreviations in the text, and they should not be introduced. Then, it is difficult (at least for me) to understand the meaning of NPM, CB, and NPC. In addition, “MBG” is used several times in the introduction, but the abbreviation is introduced only on line 71. “preE”, instead, is introduced at the beginning of introduction, but then it is repeated on lines 47 and 69. These observations are only for the introduction, I suggest revising the whole manuscript with some attention. I know that these observations may seem pedantic, but there are rules in scientific communication that should be observed and that testify the accuracy and the attention made in the preparation of a manuscript.

Response: It has been taken care of.

MBG is abbreviated on line 16 (Abstract); “Urinary marinobufagenin (MBG) is elevated in a preE rat model prior to developing hypertension and proteinuria” and in line 52-53 (Introduction); “A similar syndrome can also be triggered by daily injections of marinobufagenin (MBG) in early pregnancy [4].”

  1. Line 39-45: Two hypotheses are introduced here: they refer to hypertension-caused vasospasm and altered cerebrovascular regulation. Then endothelial dysfunction is introduced: is this a third hypothesis? What is the link with hypotheses 1 and 2?

Response: “Significant data support the theory that preE is a disorder of endothelial dysfunction associated with oxidative stress [2].” It is the third and independent hypothesis.

  1. Line 62-63: The sentence beginning with “Increased …” is a repetition of lines 42-45.

Response: “Increased vascular permeability is recognized as a key pathological event in preE [14, 6]” has been removed from line 62-63.

  1. MBG is introduced without an explanation and without an apparent reason. To also allow non-experts of preeclampsia to read and understand this research (I am not a preeclampsia expert either), I think it would be necessary to say that MBG is implicated in the pathogenesis of the disease and provide some references supporting this evidence. Overall, the introduction needs substantial rearrangement.

Response: MBG has been well introduced/explained in abstract “Urinary marinobufagenin (MBG) is elevated in a preE rat model prior to developing hypertension and proteinuria” and in Introduction “A similar syndrome can also be triggered by daily injections of marinobufagenin (MBG) in early pregnancy [4]. We have evaluated vascular leakage in this rodent "preE" model and observed greater leakage in preE rats compared to normal pregnant and non-pregnant female rats [12]. The vascular integrity was assessed by observing single post-capillary venules in the omentum, using fluorescein isothiocyanate-albumin (FITC albumin) injections to detect any extravasation of intravascular fluid [12]. Moreover, the intravenous injection of 200 nmol MBG into control rats caused the extravasation of labeled albumin within minutes. Subsequent studies on rat lung endothelial cell monolayers incubated with increasing doses of MBG demonstrated hyperpermeability, linked to the disruption of tight junctions between cells [13].

  1. Lines 153, 154, 168: You do not need to repeat the source (Sigma or Invitrogen) every time you mention the p38 or the Jnk inhibitor.

Response: It has been taken care of.

  1. Paragraph 3.1 and fig. 1: The treatment with DMSO is called “vehicle” in the text and “basal” in figure 1: please be consistent. The statistical significance is reported as p<0.001 in the text and p<0.05 in the figure legend. In the figure legend, “*p<0.05 for all MBG concentrations” is not clear (one MBG concentration does not have a significant effect): you should indicate with which value the comparison was made. In this case, it should be “*p<0.05 (or p<0.001) vs Basal (or Vehicle)”. The sentence “No significant effect …” is not necessary and should be removed. The legend of y axis says “fold-change from basal”, while in other figures it is “% of Basal”. Please be consistent.

Response: It has been taken care of. “Fold change” and “% of Basal” has been used for different figs in order to best explain the comparison of data.

  1. Fig. 2: Similar to fig.1, the text says p<0.001, while the figure legend says p<0.05. Then, “*p<0.05 for each MBG concentrations” does not mean much. It should be replaced with “*p<0.05 (or p<0.001) vs Basal (or Vehicle)”. The sentence “The 0.1 nM MBG …” is not necessary and should be removed. The sentence of line 262-263 is not clear: what does the symbol † indicate? Is it p<0.05 vs Basal? Or vs MBG (“for each concentration” is not explicative)? Is this sentence saying that the yellow and the red histograms represent values that are significantly different from both “Basal” and the corresponding MBG concentration?

Response: The sentence “The 0.1 nM MBG did not affect permeability” has been removed from the figure legends. The symbol “†” indicates attenuation of the symbol “*”.

  1. Fig. 3: see the comments to figures 1 and 2. In addition, those on the y axis are not “arbitrary units”, since each number represents the result of a ratio, and a ratio does not have a unit. But, why in this figure (and also figures 4A-C) data are not expressed as % of control, as in the other figures? The sentence “Data are shown as mean ± SEM”, with the indication of n and replicates, is not present in all figure legends. Please be consistent. Please also check all the figure legends for comments to figures 1 – 3.

Response: Response: It has been taken care of.

  1. Line 205, “control cells”: different words are used to identify controls all over the text: Basal, Vehicle, DMSO-treated, Control, Baseline. Please be consistent.

Response: It has been taken care of.

  1. Line 296: please specify p<0.05 vs …?

Response: It has been taken care of “*p < 0.05; basal and MBG-treated cells.”

  1. Figures 5B and 6: Please provide a scale bar.

Response: It will be provided during the processing/production of print version.

  1. Fig. 6: an FITC-conjugated secondary antibody has been used and observations have been made with a confocal microscope equipped with FITC filters (methods). Since FITC emits a green fluorescence, where does the red color of ZO-1, occludin, and claudin-1 come from?

Response: Texas Red's (red color) longer wavelength reduces the background fluorescence of the sample, producing a clearer, more specific image.

  1. Lines 355 and 365: what are the CTB cells?

Response: cytotrophoblast (CTB) is a layer of cells that forms the inner part of the trophoblast in a developing embryo of a placental mammal. Our  lab extensively studied the CTB cells model for the preeclampsia pathogenesis.

  1. Acknowledgments: the Authors thank dr. Mor for the CTB cell line, but these cells do not seem to have been used in these studies.

Response: Thank you very much for catching this error. This sentence has been removed from the acknowledgments.

  1. Discussion: as in the introduction, some basic information is missing, at least in my opinion, about the evidence that MBG is involved in the pathogenesis of preE. This is an essential prerequisite to justify these studies and to propose mechanistic models.

Response: The mechanistic model has been depicted in Figure 7. Working Model.

Reviewer 2 Report

Comments and Suggestions for Authors

This article evaluated the effect of marinobufagenin on human brain microvascular endothelial cells in vitro. This article is potentially interesting and beneficial to experimenter. But this article should be changed in the following main points and minor points.

Main points

It should be described whether there have been any articles of increased marinobufagenin in human cases of preeclampsia, rather than animal model. If not reported, it is necessary to state what the experimental results mean. For example, the authors created an experimental system that induced cerebral edema and demonstrated one of the phenomena that occurred when cerebral edema was induced, and it is possible that something similar may also occur in humans. It should also be stated in Introduction and Discussion (as limitation).

Authors have confirmed MBG-induced apoptosis in HBMEC treated with the p38 or Jnk inhibitors. They should explain why they didn’t show any studies using Src inhibitor.

Minor points

In the main text (from Introduction to Conclusions), the full names (abbreviations) were often repeated, or the abbreviations were suddenly written, making it difficult to read (especially MBG, pre E, HBMEC, and ECM). It will be easier for readers to read if you write out the full name of the first abbreviation you come across and then use the abbreviation consistently thereafter. Additionally, in Figure 1 through 7, each figure legend should be independent, and the first abbreviation for each figure should be replaced with the full name. Abbreviations in Figures should be specified in legend.

Page 2, line 53 and line 55 in Introduction: The authors should initially use the words in full, followed by the abbreviation in parentheses for the following words (NPM, CB, NPC, and PDS)

In References: The page numbers for each article should be written in a consistent manner.

Author Response

Comments and Suggestions for Authors

This article evaluated the effect of marinobufagenin on human brain microvascular endothelial cells in vitro. This article is potentially interesting and beneficial to experimenter. But this article should be changed in the following main points and minor points.

Main points

It should be described whether there have been any articles of increased marinobufagenin in human cases of preeclampsia, rather than animal model. If not reported, it is necessary to state what the experimental results mean. For example, the authors created an experimental system that induced cerebral edema and demonstrated one of the phenomena that occurred when cerebral edema was induced, and it is possible that something similar may also occur in humans. It should also be stated in Introduction and Discussion (as limitation).

Response: Thank you very much for your constructive advice. According a “Clinical significance” section has been added at the end of the manuscript.

  1. Clinical significance

Recent data strongly support the involvement of MBG in preE [45, 46]. Cardiotonic steroids (CTS), "endogenous digoxin-like factors" (EDLF), have been known since the 1980s to increase significantly during pregnancy-induced hypertension and preE [47,48,49]. Recently, MBG has been identified as the digoxin-like factor. MBG, but not endogenous ouabain (EO), is markedly elevated in preE [50]. We developed an ELISA with high specificity for MBG [51] which revealed a 5-fold increase in serum MBG and a 4-fold increase in urine MBG levels in preE patients vs. normotensive pregnant women [45]. Plasma levels of MBG, but not EO, become elevated in patients with preE [46]. The work is innovative because in vitro data obtained in this study evaluated the MBG-induced BBB permeability in preE. The MBG-induced pathophysiological cascade can be blocked for the treatment of preE in early human pregnancy.

Authors have confirmed MBG-induced apoptosis in HBMEC treated with the p38 or Jnk inhibitors. They should explain why they didn’t show any studies using Src inhibitor.

Response: There is no specific reason not to use the Src inhibitor. The study has been focused on p38 and Jnk pathway for the mechanism of MBG-induced apoptosis in HBMEC.

Minor points

In the main text (from Introduction to Conclusions), the full names (abbreviations) were often repeated, or the abbreviations were suddenly written, making it difficult to read (especially MBG, pre E, HBMEC, and ECM). It will be easier for readers to read if you write out the full name of the first abbreviation you come across and then use the abbreviation consistently thereafter. Additionally, in Figure 1 through 7, each figure legend should be independent, and the first abbreviation for each figure should be replaced with the full name. Abbreviations in Figures should be specified in legend.

Response: It has been taken care of.

Page 2, line 53 and line 55 in Introduction: The authors should initially use the words in full, followed by the abbreviation in parentheses for the following words (NPM, CB, NPC, and PDS)

Response: It has been taken care of.

In References: The page numbers for each article should be written in a consistent manner.

Response: It has been taken care of.

Round 2

Reviewer 1 Report

Comments and Suggestions for Authors

The manuscript has been changed and made clearer, therefore I do not have major observations. However, point 16 of my previous review remains unanswered: I do not think that the use of a certain fluorophore may increase the specificity of an immunostaining (or the specificity of an image, whatever it may mean). But this is not the point. The point is that in the methods it is said that FITC-conjugated secondary antibodies and FITC microscope filters have been used, but then images labeled with Texas red (and I supposed observed with TRITC filters) are shown. This is only a minor point, but the descriobed methods should be consistent with the data shown in the results.

Author Response

Response to the reviewer's comments.

Reviewer 1:

Comments and Suggestions for Authors

The manuscript has been changed and made clearer; therefore, I do not have major observations. However, point 16 of my previous review remains unanswered: I do not think that the use of a certain fluorophore may increase the specificity of an immunostaining (or the specificity of an image, whatever it may mean). But this is not the point. The point is that in the methods it is said that FITC-conjugated secondary antibodies and FITC microscope filters have been used, but then images labeled with Texas red (and I supposed observed with TRITC filters) are shown. This is only a minor point, but the methods described should be consistent with the data shown in the results.

Response: I am extremely sorry for confusing. Please see the statement at the figure legend in Figure 6 : “Specifically, cells treated with 1, 10, and 100 nM MBG show disrupted staining patterns with scattered ZO-1, Occludin, Claudin-1 (all red), and E-Cadherin (green).”

Reviewer 2 Report

Comments and Suggestions for Authors

This paper was precisely revised, it got better. But this article should be changed in the following a minor point. If the minor point is changed, it would be worthy of accept.

Minor point

In Figure 1 through 7, each figure legend should be independent, and the first abbreviation (MBG and HBMEC) for each figure should be replaced with the full name. Abbreviations in Figures should be specified in legend.

Author Response

Minor point

In Figure 1 through 7, each figure legend should be independent, and the first abbreviation (MBG and HBMEC) for each figure should be replaced with the full name. Abbreviations in Figures should be specified in legend.

Response: It has been taken care of (green marked).